# Generation of Porcine and Rainbow Trout 3D Intestinal Models and Their Use to Investigate Astaxanthin Effects In Vitro

**DOI:** 10.3390/ijms25115966

**Published:** 2024-05-29

**Authors:** Sharon Arcuri, Georgia Pennarossa, Rolando Pasquariello, Madhusha Prasadani, Fulvio Gandolfi, Tiziana A. L. Brevini

**Affiliations:** 1Laboratory of Biomedical Embryology, Department of Veterinary Medicine and Animal Science and Center for Stem Cell Research, Università degli Studi di Milano, 26900 Lodi, Italy; sharon.arcuri@unimi.it (S.A.); georgia.pennarossa@unimi.it (G.P.); 2Department of Agricultural and Environmental Sciences—Production, Landscape, Agroenergy, Università degli Studi di Milano, 20133 Milan, Italy; rolando.pasquariello@unimi.it (R.P.); fulvio.gandolfi@unimi.it (F.G.); 3Institute of Veterinary Medicine and Animal Sciences, Estonian University of Life Sciences, 51014 Tartu, Estonia; madhusha.gamage@emu.ee

**Keywords:** astaxanthin, 3D in vitro models, intestine, porcine, rainbow trout

## Abstract

Astaxanthin (AST) is a natural compound derived from shellfish, microorganisms, and algae, with several healthy properties. For this reason, it is widely used in the diet of humans and animals, such as pigs, broilers, and fish, where its addition is related to its pigmenting properties. Moreover, AST’s ability to reduce free radicals and protect cells from oxidative damage finds application during the weaning period, when piglets are exposed to several stressors. To better elucidate the mechanisms involved, here we generate ad hoc pig and rainbow trout in vitro platforms able to mimic the intestinal mucosa. The morphology is validated through histological and molecular analysis, while functional properties of the newly generated intestinal barriers, both in porcine and rainbow trout models, are demonstrated by measuring trans-epithelial electrical resistance and analyzing permeability with fluorescein isothiocyanate–dextran. Exposure to AST induced a significant upregulation of antioxidative stress markers and a reduction in the transcription of inflammation-related interleukins. Altogether, the present findings demonstrate AST’s ability to interact with the molecular pathways controlling oxidative stress and inflammation both in the porcine and rainbow trout species and suggest AST’s positive role in prevention and health.

## 1. Introduction

Astaxanthin (AST) is a natural lipid-soluble and red–orange xanthophyll carotenoid widely found in marine seafood, such as shrimp, lobster, and crab, especially in their shell portions, and in several microorganisms, such as Phaffia rhodozyma and certain algae [1,2,3]. AST’s capacity to regulate inflammatory responses by preventing free radical formation and modulating redox balance has been described by a number of studies carried out in different species, mostly in vivo [4,5,6,7,8,9,10,11]. Thanks to these properties, AST finds applications in pharmaceutical, nutraceutical, and cosmetic industries, with various health benefits in humans [6,12,13,14,15,16,17,18,19]. In fish, pigs, and broilers, this compound is used as a pigmenting additive, to confer on animals their distinctive and commercially valued skin and fillet color appearance, directly associated with the final product quality [4,20,21,22,23,24,25,26,27,28]. In addition, this carotenoid is a useful supplement in piglet diets, due to its ability to reduce intestinal inflammation and to boost the immune system during the weaning period, when young animals are exposed to a number of stressors [29]. In particular, Szczepanik et al. recently demonstrated that AST addition is able to reduce free radicals and protect porcine cells from oxidative damage [30]. The molecular mechanisms involved in these beneficial effects are under elucidation and, for this purpose, commercially available human (Caco-2 and HT-29) as well as pig (IPEC-J2) cell lines have been used to develop in vitro models [31,32,33,34,35,36]. In addition, all these studies have been performed using in vitro 2D culture systems that only partially reflect the complexity of the intestinal barrier. In fish, where AST is extensively used for its pigmenting properties, the information available is even more limited, due to the small number of cell lines available for in vitro studies. Based on this, we selected the pig and the rainbow trout as representative experimental models and proposed a 3D strategy that tries to mimic the complexity of the intestinal mucosae. Moreover, we co-cultured intestinal fibroblasts and epithelial cells isolated from both species onto Alvetex^®^ scaffold inserts to create functional 3D in vitro platforms. We then exposed the newly generated artificial intestinal mucosae to AST and investigated the molecular responses to the carotenoid.

## 2. Results

### 2.1. Development and Characterization of Porcine 3D Intestinal Models

Porcine intestinal fibroblasts (pIFs) and epithelial cells (pIECs) grew out of the tissue explants within 6 days of culture. They formed monolayers with cells displaying elongated morphology with small central nuclei (Figure 1A) and rounded shapes with small nuclei (Figure 1B), respectively. When fibroblasts were plated onto Alvetex^®^ scaffold inserts, they infiltrated the membrane and generated a densely populated stromal compartment (Figure 1C). In addition, picrosirius red staining demonstrated that fibroblasts released and deposited collagen (Figure 1D). Epithelial cells co-cultured on the top of the generated stromal compartment acquired a distinct polarized morphology with an elongated columnar shape (Figure 1C).

Data obtained from the molecular analysis showed a significant upregulation of transcription levels for the main stromal and epithelial-related markers, namely *VIM*, *THY1*, *VIL1*, *TJP1*, *OCLN*, and *CLDN1*, in cells grown on Alvetex^®^ scaffolds compared to those cultured in 2D monolayers (Figure 1E). Furthermore, measurements of intestinal epithelial barrier integrity by trans-epithelial electrical resistance (TEER) displayed an average of 53.8 ± 6.6 Ω·cm^2^ (Figure 1F). In parallel, permeability analysis demonstrated a reduction in dextran flux from apical to basolateral compartments of the scaffolds (Figure 1G).

### 2.2. Development and Characterization of Rainbow Trout 3D Intestinal Models

Rainbow trout fibroblasts (RTFs) cultured in 2D supports showed the typical fusiform, elongated morphology (Figure 2A), whereas proximal intestine epithelial (RTpi-MI) cells were small and rounded in shape with small nuclei (Figure 2B). When cultured on Alvetex^®^ scaffolds, RTFs created a robust stromal compartment, deposited collagen (Figure 2D), and were able to support the epithelial layer (Figure 2C). In addition, RTpi-MI cells plated on top of the RTFs displayed an elongated columnar shape and polarized morphology (Figure 2C). Expression levels of both stromal and epithelial-specific genes, namely *vim*, *vil1*, *tjp1*, *cdh1*, and *cldn3*, were increased in cells cultured in 3D systems compared to those grown on 2D culture dishes (Figure 2E). TEER values showed an average of 75.8 ± 6.6 Ω·cm^2^ (Figure 2F). Low permeability was demonstrated by a reduced dextran flux from the apical to the basolateral compartment (Figure 2G).

### 2.3. Antioxidant and Anti-Inflammatory Effects of AST on the Porcine 3D Intestinal Models

After 24 h of exposure to AST, the generated porcine 3D intestinal models significantly upregulated the expression of the main antioxidant genes analyzed, namely *GPX1* and *NADPH1* (*NQO1*), regardless of AST concentration (Figure 3). In addition, the transcription levels detected after 1 and 5 µM concentration treatments were significantly higher than the untreated (3D model) and control (DMSO) groups but statistically comparable with each other (Figure 3).

In parallel, all the tested AST concentrations induced significant downregulation of the interleukins *IL1β*, *IL6,* and *IL8* (Figure 3). In particular, 1 and 5 µM exposure resulted in a significant decrement of the gene expressions when compared to the untreated (3D model) and control (DMSO) groups but displayed statistically comparable values when analyzed with each other (Figure 3).

### 2.4. Antioxidant and Anti-Inflammatory Effects of AST on the Rainbow Trout 3D Intestinal Models

Similar to what was observed in the porcine models, after 24 h of exposure to AST, the generated rainbow trout 3D intestinal models significantly increased the expression levels of the antioxidant genes *gpx1* and *nqo1*, and downregulated the interleukins *il1β*, *il6,* and *il8*, displaying expression level trends similar to those detected in the porcine models (Figure 4).

## 3. Discussion

Three-dimensional models are currently widely used in research because of their ability to allow cells to maintain the organization of the native tissue of origin, as well as organize in a three-dimensional manner, similar to their behavior in vivo [37,38,39,40]. In this study, we generate ad hoc 3D in vitro platforms that mimic the intestinal mucosa of pigs and rainbow trout. We then take advantage of the newly created functional 3D models to investigate whether AST may interact with the molecular mechanisms possibly involved in inflammation and oxidative stress. In particular, in the present experiments, 3D cell aggregation is achieved with the use of a highly porous polystyrene scaffold, commercially available and registered under the name Alvetex^®^, which is able to maintain the cells’ original shape and encourage cell interactions [41,42,43,44,45]. This is confirmed by the results in Figure 1C and Figure 2C, which show that the formation of the stromal and the epithelial compartments in both the porcine and rainbow trout species can be appreciated. The results are also consistent with previous observations that describe an Alvetex^®^ thickness of 200 µm [42] as an ideal feature to replicate in vitro the distance of cells from blood capillaries, thus replicating the behavior of nutrients in vivo [41], and is in line with the choice of several authors that previously adopted this approach to generate body barriers in vitro, such as the intestinal mucosa [43,45], the dermis [41,42,46], and the endometrium [47,48]. Notably, all of these studies demonstrated the development of the two main barrier compartments, namely the stroma and the epithelium, similar to what is described in the present manuscript and suggested the ability of Alvetex^®^ scaffold to encourage epithelial–ECM interactions, which typically occur in vivo [49,50]. In agreement with this, the use of a 3D approach allows cells to display a more physiological phenotype than the morphology acquired by the cells in traditional monolayer culture systems, which is likely to cause unphysiological behaviors [51]. Consistent with this, the morphological analysis presented here demonstrates the ability of both fibroblasts (Figure 1A and Figure 2A) and epithelial cells (Figure 1B and Figure 2B) to rearrange on the scaffold in a robust stroma and epithelial layer, respectively, with the production of the ECM components to support the upper intestinal layer (Figure 1D and Figure 2D). Interestingly, engrafting on the scaffold results in cell functional enhancement, with an increased expression of the genes distinctive of the two cell populations. As shown in Figure 1E, *VIM* and *THY1* expression levels are significantly upregulated in porcine fibroblasts cultured on Alvetex^®^ scaffolds, compared to those grown in monolayer (Figure 1E). Similarly, Figure 2D displays a higher transcription of *VIM* in rainbow trout fibroblasts cultured in 3D. All these aspects lead us to speculate that when a robust stromal compartment is created, epithelial cells are able to better mimic in vitro the functions distinctive of their original tissue. This is consistent with the results presented in this manuscript where epithelial cells, co-cultured on top of the stromal compartment, acquire a polarized elongated columnar phenotype (Figure 1C and Figure 2C) and is in agreement with Darling et al. who demonstrated a polarized shape in 3D co-cultured epithelial cells [43]. The morphological results are, once again, confirmed by the molecular data that demonstrate a statistically significant increase in the expression of the porcine epithelial markers *VIL*, *ZO1*, *OCL1*, and *CLND* in the 3D model compared to pIECs (Figure 1E). A similar result is evident in the rainbow trout 3D intestinal model that displays a significantly higher transcription of *vil1*, *tjp1*, *cdh1*, and *cldn3* compared to RTpi-MI cells (Figure 2E). In agreement with this, Darling et al. previously demonstrated that when human epithelial cells were co-cultured with stromal cells, they exhibited a polarized phenotype [43]. Similar results were previously obtained in the porcine species [32,52,53]. However, it is interesting to note that all these authors used immortalized cell lines such as Caco-2 (human) and IPEC-J2 (pig) to generate the intestinal epithelial compartment in vitro. In our opinion, this could represent a limit since these cell lines are prone to overcrowding [54] and instability [55]. In addition, they have been previously reported to exhibit atypically high TEER and only low active transport rates, so the effect of nutritional factors cannot be reliably investigated [32]. Based on all these observations, in the present work, we used primary cells, freshly isolated from intestinal biopsies to maintain the cell’s original phenotype and, more in general, to preserve the anatomical and physiological features distinctive of the intestinal epithelial cells in vivo. As described above, in the 3D models created, the presence of the epithelial compartment is demonstrated by the active transcription of the typical tight junction markers, as well as the expression of adherens junction-related genes, suggesting the activation of a molecular expression pattern that supports barrier integrity [45]. This is in agreement with previous studies that used this strategy and showed that the formation of intercellular junctions contributed to the mechanical properties of individual epithelial cells, cell adhesion, and intercellular communication [56]. However, it is interesting to note that the barrier functionality of the newly generated 3D intestinal models is further demonstrated by the TEER values obtained in the present experiments, which are similar to those reported in vivo and indicate a resistance property close to physiological transport rates. In addition, these data well correlate and are further strengthened by the experiments assessing paracellular permeability of the newly generated barrier, which, when subjected to a 2 h exposure of 4 kDa FITC–dextran, demonstrated significantly higher retention of dextran in the apical compartment, indicating the ability of epithelial cells to prevent the paracellular flux of large molecules (Figure 1G and Figure 2G).

Exposure of the generated 3D models to AST significantly upregulates the antioxidant genes *GPX1* and *NQO1,* as well as *gpx1* and *nqo1*, in the porcine and rainbow trout species, respectively (Figure 3 and Figure 4). These results are in line with Kochi et al. who previously showed an increased expression of *Gpx1* in mouse colonic mucosa [57]. Similarly, Tian et al. reported AST’s ability to decrease aflatoxin B1-induced oxidative stress, increasing *NQO1* expression in IPEC-J2 cells [31]. The results are also consistent with Davinelli et al., who demonstrated AST’s ability to regulate, in humans, the redox-sensitive transcription factor, nuclear factor erythroid 2-related factor 2 (Nrf2), which coordinates the expression of a battery of defensive genes encoding antioxidant proteins and detoxifying enzymes [19]. Altogether, these observations indicate AST properties to counteract oxidative stress in different animals, including mice and humans, as well as the porcine and rainbow trout species. Based on the emerging evidence that oxidative stress plays a crucial role in the development and maintenance of inflammation, we investigated the effect of AST on the expression of inflammation-related transcripts such as *IL1β*, *IL6,* and *IL8* (Figure 3 and Figure 4) and observed a significant reduction, starting from a 0.5 μM concentration of the carotenoid. This response was evident both in porcine as well as in rainbow trout models and is coherent with Davinelli et al. who showed AST’s ability to modulate the nuclear transcription factor-κB (NF-κB) signaling network, with a general anti-inflammatory and antioxidative effect in mouse and rat experimental models [19].

Altogether, the present findings demonstrate the possibility of starting from primary cultures and generating 3D in vitro platforms with distinctive functional compartments that mimic the intestinal mucosa of pigs and rainbow trout. The models created represent a useful tool to investigate AST’s ability to interact with the molecular pathways controlling oxidative stress and inflammation, both in the porcine and rainbow trout species, and suggest AST’s positive role in prevention and health, partly replacing, or reducing, the use of in vivo animal models.

## 4. Materials and Methods

All reagents were purchased from Sigma-Aldrich (Milan, Italy) unless otherwise indicated.

### 4.1. Ethic Statement

Porcine fibroblasts (pIFs) and epithelial cells (pIECs) were isolated from fresh small intestine biopsies collected at the local abattoir from slaughtered adult animals. Rainbow trout fibroblasts (RTFs) and proximal intestine epithelial cells (RTpi-MI) [58] were obtained from fresh dermal and intestinal biopsies, respectively, of adult individuals collected at fish culture ponds. Fish organs were obtained from animals destined for human consumption and, therefore, were not considered animal experimentation under Directive 2010/63/EU of the European Parliament. All experiments were performed in accordance with the approved guidelines.

### 4.2. pIF Isolation, Growth, and Maintenance on Standard Plastic Dishes

Porcine intestinal explants were cut longitudinally along the connective surface and, subsequently, in small fragments of approximately 2 mm^3^. Tissues were placed on a 0.1% porcine gelatin pre-coated Petri dish (Sarstedt, Milan, Italy) and cultured in Dulbecco’s Modified Eagle Medium (DMEM, Thermo Fisher Scientific, Milan, Italy), supplemented with 20% Fetal Bovine Serum (FBS, Thermo Fisher Scientific, Milan, Italy), 2 mM of glutamine, and 2% antibiotic/antimycotic solution. After 6 days of culture, cells started to grow out and fragments were carefully removed. The pIFs were maintained in the medium described above supplemented with 10% FBS, grown in 5% CO_2_ at 37 °C, and passaged twice a week at a 1:3 ratio.

### 4.3. pIEC Isolation, Growth, and Maintenance on Standard Plastic Dishes

Porcine intestinal fragments of approximately 2 mm^3^ were cut longitudinally along the epithelial surface, transferred onto a 0.1% porcine gelatin pre-coated Petri dish (Sarstedt, Milan, Italy) and cultured in Dulbecco’s Modified Eagle Medium/Nutrient Mixture F12 (DMEM/F12, Thermo Fisher Scientific, Milan, Italy) supplemented with 5% FBS, 2 mM of glutamine, and 2% antibiotic/antimycotic solution. When cells reached confluence, they were trypsinized (Trypsin-EDTA) and transferred into a T25 flask (Sarstedt, Milan, Italy) and then maintained in 5% CO_2_ at 37 °C. The pIECs were passaged twice a week at a 1:3 ratio.

### 4.4. Development of Porcine 3D Intestinal Models

Alvetex^®^ Scaffold inserts with 12 wells (Reprocell Europe, Glasgow, UK) were prepared according to the manufacturer’s instructions. The pIFs of 0.5 × 10^6^ were seeded on inserts at days 0, 7, and 9 of culture and grown in DMEM supplemented with 20% FBS, 2 mM of glutamine, 5 ng/mL of transforming growth factor β1 (TGF β1, Thermo Fisher Scientific, Milan, Italy), 100 μg/mL of ascorbic acid, and 1% antibiotic/antimycotic solution in 5% CO_2_ at 37 °C for 14 days. The medium was refreshed twice a week. On day 14, 0.4 × 10^6^ pIECs were layered on the stromal compartment and co-cultured in DMEM/F12 supplemented with 5% FBS, 2 mM of glutamine, 1% insulin–transferrin–selenium (ITS, Thermo Fisher Scientific, Milan, Italy), 5 ng/mL of epidermal growth factor (EGF, Thermo Fisher Scientific, Milan, Italy), and 1% antibiotic/antimycotic solution for 21 days. Cultures were maintained in 5% CO_2_ at 37 °C and medium was refreshed twice a week.

### 4.5. RTF Isolation, Growth, and Maintenance on Standard Plastic Dishes

Rainbow trout dermal fragments of approximately 1 mm^3^ were placed on 0.1% gelatin pre-coated T25 flasks (Sarstedt, Milan, Italy) and cultured Leibovit’s 15 Medium (L-15, Thermo Fisher Scientific, Milan, Italy) supplemented with 20% FBS, 2 mM of glutamine, and 2% antibiotic/antimycotic solution. RTFs started to grow out of the fragments after 6 days of culture. Skin biopsies were carefully removed, and fibroblasts were maintained in the medium described above supplemented with 10% FBS. The cells were passaged twice a week at a 1:3 ratio and grown without CO_2_ at 20 °C.

### 4.6. RTpi-MI Cell Isolation, Growth, and Maintenance on Standard Plastic Dishes

Fragments of approximately 1 mm^3^ of rainbow trout intestine were placed on 0.1% gelatin pre-coated T25 flasks (Sarstedt, Milan, Italy) and cultured Leibovit’s 15 Medium (L-15, Thermo Fisher Scientific, Milan, Italy) supplemented with 10% FBS, 2 mM of glutamine, and 2% antibiotic/antimycotic solution. After 10 days of culture, cells started to grow out of the fragments, which were carefully removed. The cells were maintained in the medium described above supplemented with 5% FBS. The cells were passaged twice a week at a 1:3 ratio and grown without CO_2_ at 20 °C.

### 4.7. Development of Rainbow Trout 3D Intestinal Models

Alvetex^®^ Scaffold inserts with 12 wells were prepared according to the manufacturer’s instructions. To generate the stromal compartment, 1 × 10^6^ RTFs were seeded onto scaffolds and culture for 28 days in L-15, supplemented with 10% FBS, 2 mM of glutamine, 1% antibiotic/antimycotic solution, 5 ng/mL of TGF β1, and 100 μg/mL of ascorbic acid. After 28 days, 9 × 10^5^ RTpi-MI cells were cultured on the top of the RTFs in L-15, supplemented with 5% FBS, 2 mM of glutamine, and 1% antibiotic/antimycotic solution without CO_2_ at 20 °C for 21 days. The medium was refreshed twice a week.

### 4.8. Histological Analysis

At the end of the culture periods, the generated 3D intestinal models were fixed in 4% paraformaldehyde (PFA) for 24 h, dehydrated through a series of ethanol washes, incubated in Histoclear (Bio-optica, Milano, Italy), and embedded in paraffin. Sections 5–7 µm thick were cut, dewaxed, re-hydrated, and stained with hematoxylin and eosin (HE, Sigma-Aldrich) to visualize cell morphology, and with picrosirius red to evaluate collagen deposition. The samples were analyzed using a Leica DMR microscope (Leica Microsystems, Wetzlar, Germany).

### 4.9. Trans-Epithelial Electrical Resistance (TEER) Measurement in 3D Intestinal Models

An EVOM2 Epithelial Voltmeter with an STX3 electrode (World Precision Instrument, Friedberg Germany) was used to measure TEER of the intestinal barriers. After electrode equilibration, it was inserted into the 3 sides of each insert and the final TEER value was determined as follows:TEER (Ohm × cm^2^) = (TEER average sample insert − TEER average blank insert) × Area cm^2^

### 4.10. Permeability Evaluation of 3D Intestinal Models

Permeability was evaluated using FITC–dextran 4 kDa. An amount of 100 µg/mL of stock solution was prepared dissolving 0.0010 g of FITC–dextran (dextran, cat. No. FD4-250MG) in 10 mL of complete culture medium. The solution obtained was sterilized with a 0.22 µm syringe filter. For each well, 2 mL of culture medium was added in the basolateral compartment and 0.4 mL in the apical ones. Subsequently, 600 µL of FITC–dextran solution was added in the apical compartment. An amount of 100 µL of culture medium obtained from the basolateral compartment was collected at different time points, transferred into a 96-well plate, and analyzed using a fluorescence plate reader (Bio-Rad Laboratories, Milan, Italy).

### 4.11. AST Exposure of 3D Intestinal Models

Porcine and rainbow trout 3D intestinal models were exposed to different AST concentrations (0.5 µM, 1 µM, and 5 µM) and incubated for 24 h as previously described by Campisi et al. [59]. To this purpose, 1 mM of AST stock solution was prepared by dissolving 0.0012 g of powder in 2 mL of dimethyl-sulfoxide (DMSO). Subsequently, different working solutions were prepared. DMSO was used as a vehicle. A culture medium was used as a control (CTR).

### 4.12. Gene Expression Analysis

The TaqManGene Expression Cells-to-CT kit (Thermo Fisher Scientific, Milan, Italy) was used to extract RNA from the samples, following the manufacturer’s instructions. DNase I (1:100) was added to the lysis solution. Predesigned gene-specific primers and probe sets from TaqManGene Expression Assays (Table 1) were used for quantitative Real-Time PCR of the porcine samples. For the rainbow trout samples, primers were designed with IDT web tools (https://eu.idtdna.com/pages, accessed on 23 May 2022) using complementary DNA (cDNA) sequences of rainbow trout available on NCBI (https://www.ncbi.nlm.nih.gov/, accessed on 23 May 2022). PCR products were sequenced by Eurofins Genomics (Ebersberg, Germany, Europe) and validated through alignment with the rainbow trout transcriptome. *GAPDH* and *ACTB* were used as internal reference genes for the porcine samples, and *actb* and *ef1* were used for the rainbow trout samples. CFX96 Real-Time PCR was used as a detection system (Bio-Rad Laboratories, Milan, Italy). Target gene analysis was performed using CFX Manager software Version 3.1 (Bio-Rad Laboratories, Milan, Italy), and gene expression levels are here reported with the highest expression set to 1 and the other relative to this.

### 4.13. Statistical Analysis

Statistical analysis was performed using the Shapiro–Wilk and two-way ANOVA tests (SPSS 19.1; IBM, Armonk, NY, USA). Data were presented as mean ± standard deviation (SEM). Differences of *p* ≤ 0.05 were considered significant and are indicated with different superscripts.

## Figures and Tables

**Figure 1 ijms-25-05966-f001:**
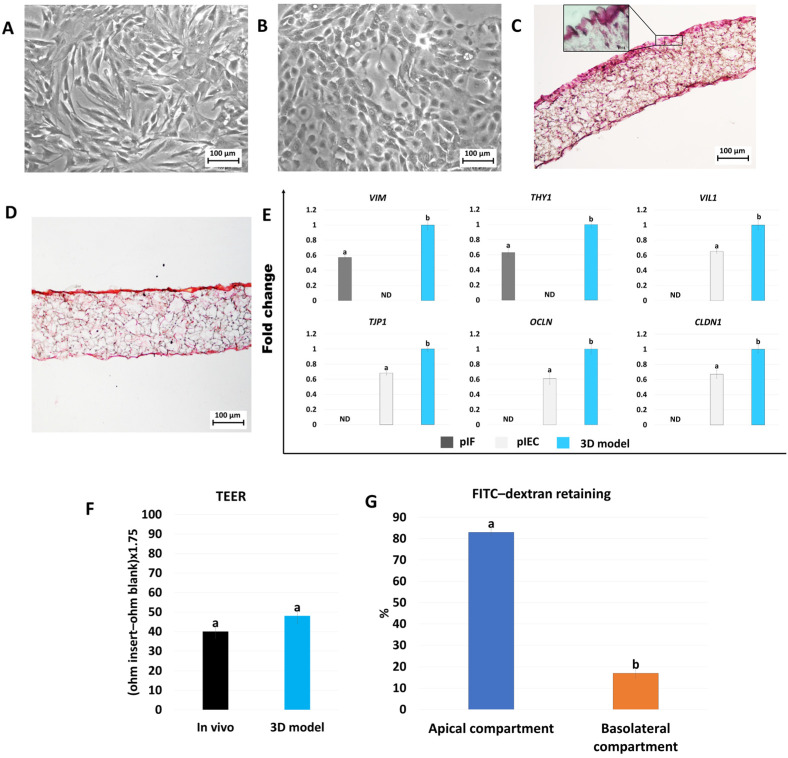
Generation and characterization of the porcine intestinal 3D models. (**A**) pIFs cultured in a 2D standard dish (scale bar 100 µm). (**B**) pIECs cultured in 2D systems (scale bar 100 µm). (**C**) Hematoxylin and Eosin staining of the 3D porcine intestinal model (scale bars 100 µm and 10 µm). (**D**) Picrosirius red staining of the 3D porcine intestinal model (scale bar 100 µm). (**E**) Gene expression levels of *VIM*, *THY1*, *VIL1*, *TJP1*, *OCLN*, and *CLDN1* genes in pIFs cultured in 2D systems (grey bars), pIECs cultured in plastic dishes (light grey bars), and 3D intestinal models obtained via co-culturing pIFs and pIECs on Alvetex^®^ scaffold inserts (3D model, sky blue bars). Data are expressed as the means ± the standard error of the means (SEM, n = 9). ^a,b^ Different superscripts indicate *p* < 0.05. (**F**) TEER values detected in native tissue (in vivo, black bar) and in in vitro 3D intestinal model (3D model, sky blue bar). Data are expressed as the means ± the standard error of the means (SEM, n = 9). (**G**) The paracellular flux of 4 kDa FITC–dextran was analyzed in apical (blue bar) and basolateral compartments (orange bar). Data are expressed as the means ± the standard error of the means (SEM, n = 9). ^a,b^ Different superscripts indicate *p* < 0.05.

**Figure 2 ijms-25-05966-f002:**
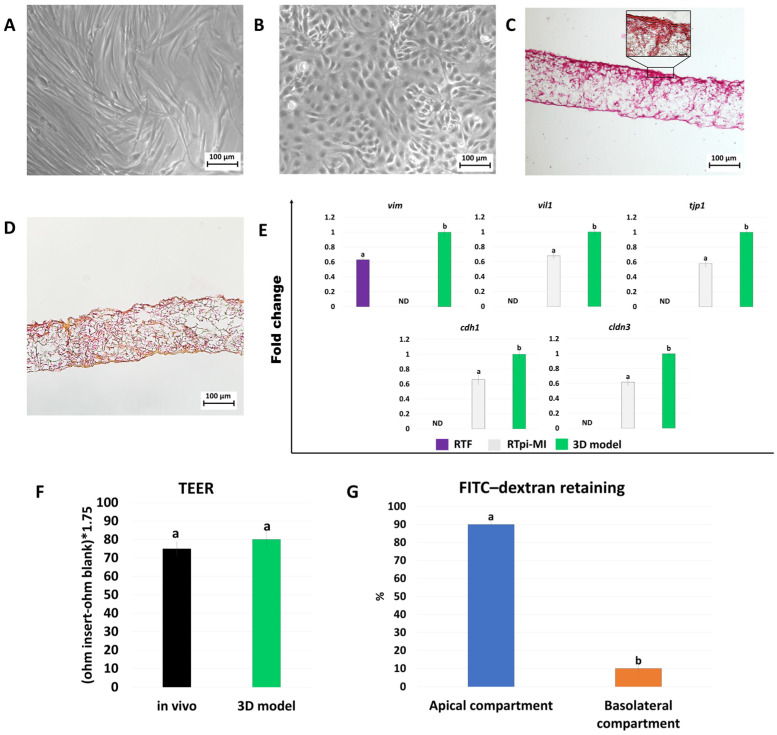
Generation and characterization of the rainbow trout 3D intestinal models. (**A**) RTFs cultured in 2D standard dishes (scale bar 100 µm). (**B**) RTpi-MI cells cultured in 2D systems (scale bar 100 µm). (**C**) Hematoxylin and eosin staining of the 3D rainbow trout intestinal model (scale bars 100 µm and 10 µm). (**D**) Picrosirius red staining of the 3D rainbow trout intestinal model (scale bar 100 µm). (**E**) Gene expression levels of *vim*, *vil1*, *tjp1*, *cdh1,* and *cldn3* genes in RTFs cultured in 2D systems (violet bars), RTpi-MI cells cultured in plastic dishes (light grey bars), and 3D intestinal models obtained via co-culturing RTFs and RTpi-MI cells on Alvetex^®^ scaffold inserts (green bars). Data are expressed as the means ± the standard error of the means (SEM, n = 9). ^a,b^ Different superscripts indicate *p* < 0.05. (**F**) TEER values detected in native tissue (in vivo, black bar) and in in vitro 3D intestinal model (3D model, green bar). Data are expressed as the means ± the standard error of the means (SEM, n = 9). (**G**) The paracellular flux of 4 kDa dextran was analyzed in the apical (blue bar) and basolateral compartment (orange bar). Data are expressed as the means ± the standard error of the means (SEM, n = 9). ^a,b^ Different superscripts indicate *p* < 0.05.

**Figure 3 ijms-25-05966-f003:**
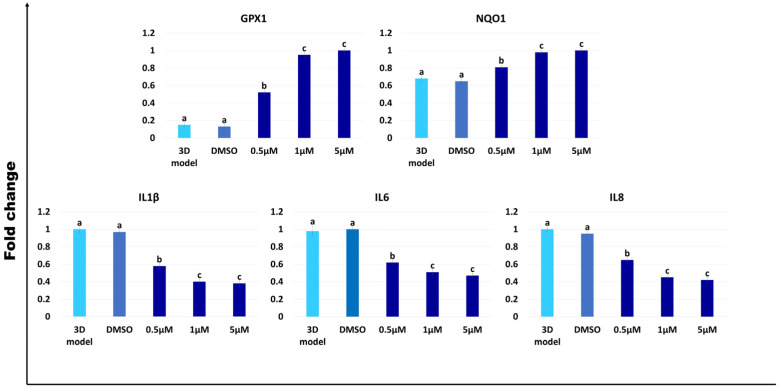
Gene expression of antioxidant (*GPX1* and *NQO1*) and inflammatory-related genes (*IL1β*, *IL6*, and *IL8*) in porcine intestinal models after different concentrations of AST exposure for 24 h. Data are expressed as the means ± the standard error of the means (SEM, n = 9). ^a–c^ Different superscripts indicate *p* < 0.05.

**Figure 4 ijms-25-05966-f004:**
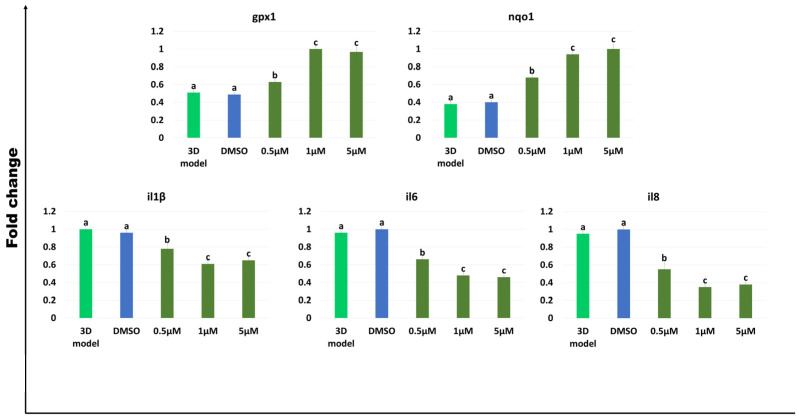
Gene expression of antioxidant (*gpx1* and *nqo1*) and inflammatory-related genes (*il1β*, *il6,* and *il8*) in rainbow trout intestinal models after different concentrations of AST exposure for 24 h. Data are expressed as the means ± the standard error of the means (SEM, n = 9). ^a–c^ Different superscripts indicate *p* < 0.05.

**Table 1 ijms-25-05966-t001:** List of primers used for quantitative PCR analysis.

Gene	Description	Cat.N./Accession No.	Species
*ACTB*	Actin, beta	Ss06827219_s1	Porcine
*CLDN1*	Claudin 1	Ss03375708_u1	Porcine
*GAPDH*	Glyceraldehyde-3-phosphate dehydrogenase	Ss03373052_u1	Porcine
*GPX1*	Glutathione peroxidase 1	Ss03383336_u1	Porcine
*IL1* *β*	Interleukin 1, beta	Ss03821899_s1	Porcine
*IL6*	Interleukin 6	Ss03394904_g1	Porcine
*IL8*	Interleukin 8	Ss03392437_m1	Porcine
*NADPH1*	NAD(P)H quinone dehydrogenase 1	Ss04246167_m1	Porcine
*OCLN*	Occludin	Ss06867496_m1	Porcine
*THY1*	Thy-1 cell surface antigen	Ss03376963_u1	Porcine
*TJP1*	Zonula Occludens 1	Ss03373514_m1	Porcine
*VIL1*	Villin 1	Ss06886976_m1	Porcine
*VIM*	Vimentin	Ss04330801_gH	Porcine
*actb*	Actin	NM_001124235	Rainbow Trout
*ef1*	Elongation factor 1	NM_001124339.1	Rainbow Trout
*vil1*	Villin 1	XM_021579239	Rainbow Trout
*tjp1*	Zonula Occludens 1	XM_021607172.1	Rainbow Trout
*cldn3*	Claudin 3	XM_021587920	Rainbow Trout
*cdh1*	e-Cadherin	XM_021607117	Rainbow Trout
*gpx1*	Glutathione peroxidase 1	NM_001124525.1	Rainbow Trout
*nadph1*	NAD(P)H quinone dehydrogenase 1	XM_021561062.2	Rainbow Trout
*il1* *β*	Interleukin 1, beta	XM_036979104.1	Rainbow Trout
*il6*	Interleukin 6	NM_001124657.1	Rainbow Trout
*il8*	Interleukin 8	NM_001124362.1	Rainbow Trout

## Data Availability

The data presented in this study are available upon request from the corresponding author.

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
