# Peer review of "Generation of Porcine and Rainbow Trout 3D Intestinal Models and Their Use to Investigate Astaxanthin Effects In Vitro"

_ijms, 2024, doi:10.3390/ijms25115966_

Round 1
Reviewer 1 Report
Comments and Suggestions for Authors
The paper’s topic is developing porcine and rainbow trout 3D intestinal models and using them to investigate the in vitro effects of astaxanthin.
The paper is well-written and contains some novelties. The development of the 3D intestinal model is based on previously published methods with some modifications in the porcine model, but it has more novelties for rainbow trout.
There are remarks which require modification in the text in order of appearance:
L 40-41: The health benefit of cosmetics in human health is questionable.
L 41-43: Astaxanthin is not being used as a pigmenting additive in pigs.
L 46: It's essential to note that reference [20] is not directly related to the effect of carotenoids, which is a key aspect of your research. Ensuring the relevance of your references is vital for the credibility of your work.
L 114: Abbreviations of gene names should be written in Italics.
L 117 & 121: When presenting data as mean ± SEM, including the number of samples analysed, it is important. This detail is crucial for the comprehensiveness and reliability of your findings.
L 162: Abbreviations of gene name abbreviations should be written in Italics.
L 165 & 169: The number of samples analysed should be given if the data show the mean ± SEM.
L 174: The changes in the expression of GPX1 are remarkable, but this gene encodes the cytosolic GPx1. GPX2 would be more specific in an intestinal model, but GPX2 encodes it.
L 175: The official name of the gene NADPH1 should be given in brackets at the first appearance of NOQ1 because the first one is given in Table 1.
L 200: The number of samples analysed should be given if the data show the mean ± SEM.
L 233: The number of samples analysed should be given if the data show the mean ± SEM.
L 240: ’In this study’ instead of ’in this manuscript.’
L 243-244: The study's results showed the dose-dependent effect of AST on the gene expression of some antioxidant and inflammatory response genes. However, no direct evidence exists that AST controls inflammation and oxidative stress. The statement would be correct if oxidative stress or inflammatory processes were induced in the model.
L 274: Please add the number of references to this statement, not only the first author's name.
L 307: Abbreviations of the gene name abbreviations should be written in Italics. Additionally, abbreviations of the gene names should be given in capital letters for porcine and small letters for fish.
L 308-309: The cited reference does not support the results of the present study because it was an azoxymethane (AOM)-induced model, and the gene expression of NQO1 was not measured.
L 336-340: Sampling from swine in the abattoir is acceptable according to the guidelines, but a more detailed description is required for the sampling from rainbow trout. The statement that fish samples collected at fish culture ponds is too general.
Table 1: Please mention and show accurately which were porcine and rainbow trout genes. Otherwise, you should explain the reason for the difference: three housekeeping genes were used for porcine gene expression analysis and one for rainbow trout gene expression analysis.
Author Response
The paper’s topic is developing porcine and rainbow trout 3D intestinal models and using them to investigate the in vitro effects of astaxanthin.
The paper is well-written and contains some novelties. The development of the 3D intestinal model is based on previously published methods with some modifications in the porcine model, but it has more novelties for rainbow trout.
We thank the Reviewer for her/his appreciation of our study and for the comments and suggestions that will help improving the quality of the manuscript.
There are remarks which require modification in the text in order of appearance:
L 40-41: The health benefit of cosmetics in human health is questionable.
AST benefits in human health have been widely demonstrated, especially in pharmaceutical and nutraceutical industries. We agree with the Reviewer that its use in cosmetics is more limited. However, based on evidence dating from 2018 onwards, AST use in the cosmetic field, as anti-aging, anti-UV irradiation, wound healing, photoprotective, antioxidant and anti-inflammatory, is being rapidly implemented. To support this aspect, we added references number 12-19. Please see lines 41 and 597-614.
L 41-43: Astaxanthin is not being used as a pigmenting additive in pigs.
We agree with the Reviewer that AST is not routinely used as additive in pigs. However, Carr et al. demonstrated that loin chops, obtained from animals fed with a diet supplemented with AST, acquired a darker colour than those fed without AST. This reference was added in the text. Please see lines 43 and 641-643.
L 46: It's essential to note that reference [20] is not directly related to the effect of carotenoids, which is a key aspect of your research. Ensuring the relevance of your references is vital for the credibility of your work.
We apologize for this mistake. Reference 20 was removed.
L 114: Abbreviations of gene names should be written in Italics.
Gene names are now written in Italics.
L 117 & 121: When presenting data as mean ± SEM, including the number of samples analysed, it is important. This detail is crucial for the comprehensiveness and reliability of your findings.
The number of samples analysed was added in the text. Please see lines 126, 128, 130
L 162: Abbreviations of gene name abbreviations should be written in Italics.
Gene names are now written in Italics.
L 165 & 169: The number of samples analysed should be given if the data show the mean ± SEM.
The number of samples analysed was added in the text. Please see lines 174, 176, 178.
L 174: The changes in the expression of GPX1 are remarkable, but this gene encodes the cytosolic GPx1. GPX2 would be more specific in an intestinal model, but GPX2 encodes it.
GPX1 and GPX2 are known to be the two major enzymes that reduce hydroperoxides in intestinal epithelium (Chu et al., https://doi.org/10.1158/0008-5472.CAN-03-2272). In this manuscript we used GPX1, but we fully agree with the Reviewer that extending the gene panel to GPX2 as well will be a further improvement for future studies.
L 175: The official name of the gene NADPH1 should be given in brackets at the first appearance of NOQ1 because the first one is given in Table 1.
The text was modified as suggested. Please see line 184.
L 200: The number of samples analysed should be given if the data show the mean ± SEM.
The number of samples analysed was added in the text. Please see line 209.
L 233: The number of samples analysed should be given if the data show the mean ± SEM.
The number of samples analysed was added in the text. Please see line 242.
L 240: ’In this study’ instead of ’in this manuscript.’
The sentence was modified according to the Reviewer’s suggestion.
L 243-244: The study's results showed the dose-dependent effect of AST on the gene expression of some antioxidant and inflammatory response genes. However, no direct evidence exists that AST controls inflammation and oxidative stress. The statement would be correct if oxidative stress or inflammatory processes were induced in the model.
We thank the Reviewer for this observation. We modified the statement to clarify this point. Please see lines 251-253.
L 274: Please add the number of references to this statement, not only the first author's name.
The number of references was added. Please see line 302.
L 307: Abbreviations of the gene name abbreviations should be written in Italics. Additionally, abbreviations of the gene names should be given in capital letters for porcine and small letters for fish.
Gene names are now written in Italics.
L 308-309: The cited reference does not support the results of the present study because it was an azoxymethane (AOM)-induced model, and the gene expression of NQO1 was not measured.
We apologize for this. The sentence was reformulated and extended in agreement with the Reviewer’s suggestion. Please see lines 331-333.
L 336-340: Sampling from swine in the abattoir is acceptable according to the guidelines, but a more detailed description is required for the sampling from rainbow trout. The statement that fish samples collected at fish culture ponds is too general.
Detailed information were added in the text (please, see Lines 390-392)
Table 1: Please mention and show accurately which were porcine and rainbow trout genes. Otherwise, you should explain the reason for the difference: three housekeeping genes were used for porcine gene expression analysis and one for rainbow trout gene expression analysis.
We modified the table and the related text adding the requested details.
Reviewer 2 Report
Comments and Suggestions for Authors
In the current experiment, the authors assessed how astaxanthin affects the morphology of the intestine. To do so, they used an experimental platform based on a 3D model of the pig intestine and rainbow trout gut cultured from fibroblasts. The work is undoubtedly interesting, but as it is only based on an in vitro model it has limited cognitive significance. I have several doubts about the methodology that the authors should clarify.
Critical comments
336 - From which segment of the intestine the fibroblasts were taken. It is not clear at all whether it was the small intestine or the large intestine.
Line 340 - please provide the Bioethics committee approval number for this experiment.
Line 343 - it is unclear how the authors avoided contamination of the samples with fungi.
Line 404 - what parameters were assessed during the histological analysis?
Line 435 - please describe how the primers were designed and validated.
Line 445 - what test was used to assess normal distribution? What two factors were considered before the two-way ANOVA?
Author Response
In the current experiment, the authors assessed how astaxanthin affects the morphology of the intestine. To do so, they used an experimental platform based on a 3D model of the pig intestine and rainbow trout gut cultured from fibroblasts. The work is undoubtedly interesting, but as it is only based on an in vitro model it has limited cognitive significance. I have several doubts about the methodology that the authors should clarify.
We thank the Reviewer for finding our work interesting.
Critical comments
336 - From which segment of the intestine the fibroblasts were taken. It is not clear at all whether it was the small intestine or the large intestine.
Fibroblasts were isolated from the small intestine. Details were added in the text. Please see line 386.
Line 340 - please provide the Bioethics committee approval number for this experiment.
Organs were isolated from animals destined to human consumption and, therefore, were not considered as animal experimentation under Directive 2010/63/EU of the European Parliament. Details were added in the text. Please see lines 390-392.
Line 343 - it is unclear how the authors avoided contamination of the samples with fungi.
We apologise for this inaccuracy. Fungi contamination was preventive since all isolation was always carried out in medium supplemented with 2% antibiotic/antimycotic solution. All culture procedures were performed in medium containing 1% antibiotic/antimycotic solution. Details were added in the text.
Line 404 - what parameters were assessed during the histological analysis?
Sections were stained with haematoxylin/eosin to visualize cell morphology, and with Picrosirius red to evaluate collagen deposition.Details were added in the text. Please see lines 463-464.
Line 435 - please describe how the primers were designed and validated.
We apologise for missing this description. Details were added in the text. Please see lines 504-509.
Line 445 - what test was used to assess normal distribution? What two factors were considered before the two-way ANOVA?
Statistical analysis was performed using Shapiro-Wilk test and two-way Anova considering the time of exposure and AST concentration as the two factors. The text was modified. Please see lines 542-543.